# Structural Imaging Characteristic, Clinical Features and Risk Factors of Cerebral Venous Sinus Thrombosis: A Prospective Cross-Sectional Analysis from a Tertiary Care Hospital in Pakistan

**DOI:** 10.3390/diagnostics11060958

**Published:** 2021-05-26

**Authors:** Safia Bano, Muhammad Umer Farooq, Sarwat Nazir, Ayesha Aslam, Adnan Tariq, Muhammad Athar Javed, Habib Rehman, Ahsan Numan

**Affiliations:** 1Department of Neurology, King Edward Medical University, Lahore 54000, Pakistan; safiabano207@gmail.com (S.B.); dr.umer7035@gmail.com (M.U.F.); drayesha_azlam15@yahoo.com (A.A.); adnan679@gmail.com (A.T.); dratharjaved59@gmail.com (M.A.J.); 2Department of Gynecology and Obstetrics, Fatima Jinnah Medical University, Lahore 54000, Pakistan; sarwatahsan1@hotmail.com; 3Department of Physiology, University of Veterinary and Animal Sciences, Lahore 54000, Pakistan; habibrehman@uvas.edu.pk

**Keywords:** venous sinus, parenchymal lesions, stroke, headache, pregnancy, neurological disorders

## Abstract

Cerebral venous sinus thrombosis (CVST) is a rare cause of stroke that accounts for 0.5–1.0% of all strokes. Clinical presentation, predisposing factors, neuroimaging findings, and outcomes of CVST are extremely diverse, which causes a high index of suspicion in diagnosis. Therefore, early diagnosis of CVST is crucial for prompt treatment to prevent morbidity and mortality. Objective: The purpose of this prospective study is aimed at assessing the clinical characteristics, potential risk factors, and neuro-radiological features along with the topography of venous sinus involved in CVST patients in a tertiary care hospital, Lahore, Pakistan. Material and Methods: Consecutive patients enrolled in this study had a computed tomography (CT) scan, magnetic resonance imaging (MRI), and magnetic resonance venography (MRV) along with a clinical presentation to confirm the diagnosis of CVST. Categorical data were presented as percentages. Continuous variable and categorical data were compared (parenchymal lesions vs. non-parenchymal lesions) using the Student’s t-test and Chi-square test, respectively. Results: A total of 3261 patients with stroke were presented during the study period. Out of all patients, 53 confirmed patients with CVST (1.6%) were recruited; the predominant population was female (84.91%), having a male to female ratio of 1:4. Mean age of the cohort was 28.39 ± 7.19 years. Most frequent symptoms observed were headache (92.45%) followed by vomiting (75.47%), seizures (62.26%), papilledema (54.72%), visual impairment (41.51%), and altered consciousness disturbance (52.83%). The presumed risk factors associated with CVST were puerperium (52.83%), use of oral contraceptives (13.21%), antiphospholipid syndrome (7.55%), elevated serum levels of protein C and S (5.66%), and CNS infection (3.77%). On cranial CT scans, 50 patients (94.33%) showed abnormalities while 32 patients exhibited various parenchymal lesions. Seizures were more frequent in CVST patients with parenchymal lesions compared with subjects lacking parenchymal lesions. Seventy-two sinuses, either single or in combination, were involved in CVST patients, being more common in patients with parenchymal lesions than those without parenchymal lesions. The most frequent locations of CVST were the superior sagittal and transverse sinus. Conclusion: In short, non-contrast CT brain may be used as a first line investigation in suspected cases of CVST. Our study also demonstrates some regional differences in the clinical features, risk factors, and neuroimaging details of CVST as described by some other studies. Therefore, care must be taken while diagnosing and predicting the outcome of the CVST.

## 1. Introduction

Cerebral venous sinus thrombosis (CVST), thrombosis of the dural sinus and/or cerebral veins, is a rare form of stroke, accounting for 0.5–1% of all strokes, and often affects the young population [1]. CVST is considered a rare neurovascular disorder, occurring in 3–4 per million in the West [2,3]. However, the incidence of CVST has been reported to be higher in South-East Asia during the puerperal period [4]. A six-year, in-hospital imaging prevalence of CVST was found to be 11.05% in Pakistan [5]. About 20% of strokes related to CVST have been reported in young Asian women [6]. Another study from India also reported that CVST accounted for half of the young stroke cases and 40% of strokes in women [7]. Various predisposing factors shown to be associated with CVST include pregnancy and puerperium, malignancy, oral contraceptives, thrombophilias, systemic inflammatory diseases, nephrotic syndrome, chronic inflammatory diseases, tuberculosis, meningitis, and dehydration that may be present either alone or in combinations in 85% of the cases [8,9]. Clinical presentation of CVST is highly variable and symptoms have been found to range from isolated headache to focal neurological symptoms and signs, seizures, and coma [10]. Due to the wide spectrum of presenting symptoms, clinical suspicion is very important for an accurate diagnosis of CVST [11]. A multicenter study involving Pakistan and the United Arab Emirates demonstrated that headache (81%), focal motor deficits (45%), seizures (39%), and mental status changes (37%) were more frequently presenting features in 109 patients with cerebral venous thrombosis [12]. On structural imaging, the involvement of cerebral sinus has been reported in the majority of patients with CVST [13].

Variation in the clinical presentations renders the clinical diagnosis of CVST more difficult and, sometimes, invites a high index of suspicion in diagnosis. Additionally, there is a possibility of regional variation regarding the frequency of risk factors and clinical presentations [14]. A recent study regarding CVST in Asian countries including Pakistan showed differences in the clinical presentations and predisposing factors in comparison with the USA and other Western countries [15]. Due to the higher prevalence and mortality rate in developing countries compared with developed countries, it becomes imperative to further clarify clinical presentation and rapid neuroimaging along with identification of all presumed risk factors associated with CVST so that early diagnosis may be made in a clinical setup. Therefore, the purpose of this prospective cross-sectional, single-center study was to characterize the clinical characteristics, potential risk factors, and neuroradiological features of patients with CVST in a tertiary care hospital located in Lahore, Pakistan.

## 2. Materials and Methods

### 2.1. The Study Design

A prospective cross-sectional, single-center study was conducted from January 2017 to December 2019 of all the patients admitted for stroke at the Department of Neurology, King Edward Medical University, Lahore, Pakistan (a tertiary care hospital). CVST was diagnosed using the established diagnostic criteria [16] that include (i) clinical hypothesis of CVST (headache, focal neurological deficits, and cranial hypertension) (ii) supported by neuroimaging showing a “delta sign” on contrast cranial computed tomography (CT) scan, and (iii) magnetic resonance imaging (MRI) or MR venography demonstrating cerebral sinus or venous occlusion. Cranial hypertension was diagnosed clinically followed by examination of cerebrospinal fluid collected through lumbar puncture and neuroimaging. CT angiography was not performed in every case because clinical symptoms and signs and neuroimaging were highly suggestive of CVST. After diagnosis, all consecutive patients (20–60 years old) with confirmed CVST were included in the study after getting written consent.

### 2.2. Exclusion Criteria

Patients were excluded from the study who had a history or clinical symptoms suggestive of arterial stroke or primary intracerebral hemorrhage, were younger than 18 years and older than 60 years, had a paranasal sinus infection, impaired vision by optic nerve papilledema as identified with fundoscopy by a qualified ophthalmologist, or non-contrast CT showing arterial territory infarct or hemorrhage.

### 2.3. Clinical Evaluation

Data collected from the study were demographic characteristics (age, gender, and marital status), medical, and drug history. Clinical features of patients (headache, seizures, focal weakness, visual impairment and diplopia, presence of altered sensorium, somnolence, and cognitive impairment) were recorded. Time of onset of CVST was categorized as acute (less than 48 h), sub-acute (48 h to 30 days), and chronic (more than 30 days) as described earlier [17]. The potential predisposing factors associated with CVST—including pregnancy, infections, deep vein thrombosis, oral contraceptives, malignancies, history of nephrotic syndrome, jaundice, paranasal infection, fever, history of hypertension, diabetes mellitus, and dyslipidemia—were also recorded.

### 2.4. Magnetic Resonance Imaging (MRI) and Magnetic Resonance Venography (MRV)

Neuroradiological imaging was performed by experienced radiologists, who were not aware of the clinical symptoms and signs of patients to exclude bias in the diagnosis. Magnetic resonance imaging (MRI) was performed on a 1.5 T machine (Sigma GE Medical System, Wisconsin). Fast spin Echo T2, T1, and DW images were recorded in the coronal and axial planes. Location, extent, and nature of abnormalities were recorded. The magnetic resonance venography (MRV) 2D-TOF/SPGR angiographic technique was used in both the coronal and axial planes, with an inferior saturation band to eliminate signals from arterial structures. Sections with a thickness of 1.5 mm were acquired in a coronal plane using the following parameters: 24/4.9 (TR/TE), 50-degree flips angle, 210 × 165 mm field of view, 256 × 128 matrix, and 1 NEX. Because of frequent normal variation in the transverse sinus, thrombosis was diagnosed only when a T1 hyperintense signal on MRI corresponding to a blood clot was obtained or there was evidence of collaterals. The MRV source images thus obtained were post-processed generating 12 maximum intensity projections at 15-degree increments.

### 2.5. Statistical Analysis

Data were processed in SPSS Statistics Version 20.0 (SPSS Inc., Chicago, IL, USA). Means and standard deviations were calculated for quantitative variables (age). Categorical data (gender, clinical features, risk factors, and sinuses involved) were presented as percentages. Chi-square was also employed to determine the frequency of occurrence of various categorical variables of clinical presentation, onset of disease, and proposed risk factors among recruited patients. Means of age and percentages of categorical data were compared between two groups (parenchymal lesions vs. non-parenchymal lesions) using the Student’s *t*-test and Chi-square test, respectively. The level of significance was predetermined at *p* < 0.05.

## 3. Results

Of 3261 patients presenting with stroke to the department during the study period, only 53 patients (1.6%) fitted the inclusion criteria. Mean age of patients was 28.39 ± 7.19 years (range 20–53 years). Females (*n* = 45; 84.91%) were the predominant population (mean age 28.35 ± 7.49 years, range 20–53 years) compared with males (*n* = 8; 15.09%, mean age 28.62 ± 5.62 years, range 20–34 years). Demographic data, clinical features, and presumed risk factors of recruited patients are shown in Table 1.

### 3.1. Clinical Presentations

The most common symptoms observed in our cohort were headache in 49 patients (92.45%), followed by vomiting in 40 (75.47%), seizures in 33 (62.26%), papilledema in 29 (54.72%), visual impairment in 22 (41.51%), and consciousness disturbance in 28 (52.83%) patients. Weakness was more expressed as left-side weakness compared with the right side (Table 1). As shown in Table 1, the onset of CVST in the study was sub-acute in most of the cases (79.25%).

### 3.2. Risk Factors

Table 1 shows that the most common risk factors or potential causes of CVST in descending order were puerperium (52.83%), oral contraceptives (13.21%), antiphospholipid syndrome (7.55%), elevated serum levels of protein C and S (5.66%), and CNS infection due to sinus (3.77%). In addition, we also found that seizures occurred more frequently (*p* < 0.05) in patients with parenchymal lesions compared with the patients without parenchymal lesions (Table 2).

### 3.3. Neuroradiological Findings

Out of 53 patients, 3 (5.7%) patients showed a normal CT scan whereas abnormalities were found in all 53 patients when assessed on MRV. Of 50 patients showing abnormal cranial CT findings, 32 (64%) patients were presented with focal parenchymal abnormalities that included hemorrhage/mixed density lesions in 23 (71.8%), focal edema in 5 (15.6%), and infarct not following vascular territory in 4 (12.5%) patients. In patients with non-parenchymal lesions (*n* = 18), delta signs were observed in 15 patients along with delta and cord signs in 3 patients. On MRV, a total of 72 sinuses were found affected in 53 patients (Table 3). The distribution by a number of sites showed that most of patients had a single location (*n* = 33), 15 patients had two locations, and only 2 patients had three locations. The details of the frequency and distribution by the site are depicted in Table 3. Neuroradiological imaging showed that the most common sinus affected was superior sagittal sinus followed by transverse sinus, independent of parenchymal lesions (Table 3). Similarly, the numbers of sinuses involved were more (*p* < 0.001) in patients with parenchymal lesions compared with the patients having non-parenchymal lesions (Table 3). Further details of abnormalities in the sinus are also shown in Figure 1.

## 4. Discussion

Cerebral venous sinus thrombosis (CVST) is not uncommon in Asia, especially in the South Asian subcontinent. Unlike an arterial stroke, CVST occurs more frequently in young individuals [8]. Patients enrolled in our study also had CVST at a much younger age (mean age 28years) compared to the International Study on Cerebral Vein and Dural Sinus Thrombosis (ISCVT), the largest study of the disease [9], Cerebral Vein Thrombosis International Study (CEVETIS) [18], and Cerebral Venous Sinuses Thrombosis (VENOST) [19], ranging from 39 to 41 years. However, this age was comparable to those in other studies conducted in Asian countries where ages ranged from 31 to 32 years [15,20]. The frequency of CVST was more in women (86.7%) in our cohort compared with 74.5% in ISCVT [9], 74% in CEVETIS [18], 68% in VENOST [19], and 59% in Asian Study of Cerebral Venous Thrombosis (ASCVT) [15].

One of the main challenges in the diagnosis of CVST is varying and misleading clinical presentation. Headache was the most common presenting symptom observed in our study, accounting for 92%. Headache, usually reflecting an increase in intracranial pressure, is the most common symptom of CVST and was found in >90% of patients in the ISCVT [9], in India [13], and Turkey [10]. However, in another study in Pakistan, the frequencies of headache (26%) and weakness (23%) were quite low compared to those in our patients [5]. Thirty-three patients (62.2%) in our study had seizures, which was quite high when compared with others [5,21,22]. These differences might be due to the site of involvement of cerebral sinus. Another explanation of this discrepancy of the high frequency of altered CT, together with the high frequency of symptoms expressing raised intracranial pressure (headache, seizures, vomiting, altered consciousness, papilledema), might also be due to a latency in diagnosing/presenting as quite a number of patients were also refereed by the secondary health units located outside the area of the present study.

It has been found that the frequency of seizures was more (*p* < 0.05) in CVST patients that had involvement of cortical sinuses compared with deep cerebral sinuses [22]. In addition, we also found that seizures occurred more frequently in patients with parenchymal lesions compared to patients without parenchymal lesions (71.8% vs. 38.8%). Our finding is in line with a Chinese cohort where the frequency of seizures was more than double in CVST patients having parenchymal lesions than in patients with no parenchymal lesions [23]. It is inferred that patients with parenchymal lesions have a higher risk of seizures. Therefore, CVST patients with parenchymal lesions and showing seizures may be prescribed with antiepileptic agents during a first attack. On the other hand, we are not sure about the use of anticonvulsant therapy for CVST patients presenting without any seizures but having parenchymal lesions, due to controversy in the literature [24,25].

Papilledema on fundoscopy was identified in 54.7% of patients, which is quite higher compared to others, for example, the figure was 28.3% in the ISCVT study [9], and 12.5% in Italy [22]. Another study involving a cohort of Pakistan and United Arab Emirates patients demonstrated that 35% of CVST patients had papilledema [12]. Though the higher frequency of papilledema may be suggestive of more parenchymal involvement and longer duration of thrombosis, we could not find this in our study as papilledema was equally distributed in patients with or without parenchymal lesions (Table 3). Nevertheless, this presentation could be an important indicator for imaging in resource-limited settings and care must be taken by a clinician for accurate diagnosis of CVST if a patient presents with a combination of headache, seizures, and papilledema. The signs and symptoms may vary depending on the topography of venous thrombosis, individual variation in cerebral venous anatomy, and parenchymal lesions.

In our study, we identified notable differences in presumed risk factors when compared with European, North American, and Asian studies. In the present study, the gender-specific etiology of CVST was associated with pregnancy and postpartum in 31/45 (68.8%) patients and oral contraceptive use in 7/45 (15.5%) patients. Most patients in our study were not only young but more frequently were females, being pregnant/in the puerperium period. There was a marked variation in the frequency of pregnancy/puerperium when compared with other studies. For instances, pregnancy/puerperium was found to be a potential risk factor of CVST and reported to be 18% in the ASCVT [15], 27 % in VENOST [19], 8% for CEVETIS [18], 19% for ISCVST [9], 10% in NIZAM [21], 23.4% in the North Indian population [14], and 34% in Turkey [10]. Oral contraceptives increase the risk of developing CVST in women of reproductive age [26]. In Pakistan and other Asian countries, use of contraceptive pills is uncommon. Our results on oral contraceptive pills as potential risk factors (16.2%) are quite comparable with those of other studies, which found a 16% risk in a Turkish cohort [10] and 13.3–16.6% in India [27,28], but the risk was very low when compared with results of the ISVCT (54%) [9], 39% for CEVETIS [18], 31% for Australia [29], and 47.7 % in the Italian population [22].

Neuroimaging is generally considered as a principal basis for the diagnosis of CVST. The topography and extension of CVST have rarely been investigated. Topographically, the thromboses of the superior sagittal sinus and transverse sinus were the most frequent in our study. In the study, 36 patients (67.9.6%) had single sinus involvement, 15 (28.3%) had double sinus involvement and only 2 patients (3.7%) had multiple-sinus involvement. Our results are inconsistent with frequency by topography data from the various studies, though the proportion of involvement of various sinuses was quite variable as this may depend upon the type and timing of neuroimaging [14,23,29,30]. We also demonstrated that the total number of sinuses involved was more (*p* < 0.001) in CVST patients showing parenchymal lesions than those without parenchymal lesions (86.1% vs. 31.8%). However, others could not find any difference in the frequency of involvement of various sinuses in CVST patients with or without parenchymal lesions [23]. It is pertinent to mention that treatments during admission and on discharge are not reported here because the present study is the first phase of a project designed to investigate the occurrence of CVST and also evaluation of various treatment regimens. In the present paper, only a description of basal data is reported, while treatment and follow-up will be reported later.

## 5. Conclusions

Our cohort revealed that headache was the most frequent presentation of CVST patients followed by vomiting, seizures, consciousness disturbances, and papilledema. However, the frequencies were quite high compared with other studies. The main predisposing factor identified was pregnancy/puerperium. Superior sagittal sinus and transverse sinus (both left and right) were among the most frequently involved. CT brain plain is a non-invasive, sensitive modality and readily available neuro-imaging technique in emergency settings. Non-contrast CT brain can be used as a first line investigation in suspected cases of CVST along with notable clinical features (headache and seizure ) and commonly identified risk factors (young age and puerperium state) in the proper early diagnosis of CVST.

The results of this clinical study must be interpreted very cautiously in light of several important limitations. Despite being prospective in nature, this was a single center study extended for a limited time period. As we recruited only 53 patients, care therefore must be taken when comparing the results of this study with other population-based studies involving larger cohorts, such as the ISCVT [9], CEVETIS [18], and VENOST [19] or ASCVT [15].

## Figures and Tables

**Figure 1 diagnostics-11-00958-f001:**
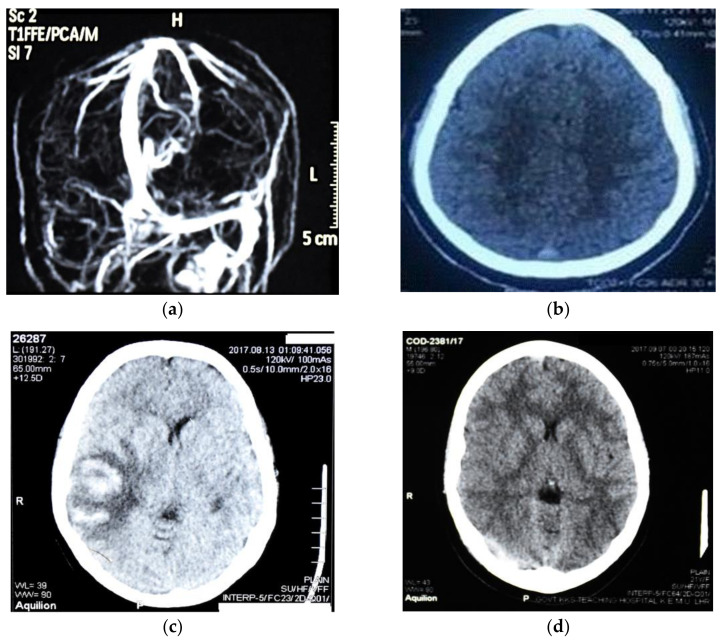
Imaging details of the complications presented in CVST patients: (**a**) sagittal contrast enhanced MRV showing filing defect in right transverse and sigmoid sinus; (**b**) CT scan brain plain showing delta sign suggestive of superior sagittal sinus thrombosis; (**c**) CT scan brain plain showing mixed density lesion not following vascular territory suggestive of transverse sinus thrombosis; (**d**) non-enhanced CT showing hyper-dense transverse sinus.

**Table 1 diagnostics-11-00958-t001:** Demographic data, clinical characteristics, and risk factors of patients (*n* = 53).

Characteristics	Frequency *		Probability
	Number	%	
Demographic data			
Age (mean ± SD) years	28.39 ± 7.19	-	-
Female sex	45	84.9	-
Married	46	86.79	-
Clinical presentation			*p* < 0.001 (df 6)
Headache	49	92.45	
Vomiting	40	75.47	
Altered state of consciousness	28	52.83	
Blurred vision	22	41.51	
Seizures	33	62.26	
Papilledema	29	54.72	
Weakness	25	47.17	
Left side **	15/25	60.00	
Right side **	10/25	40.00	
Onset of disease	-	-	*p* < 0.001 (df 2)
Acute	8	15.09	
Sub-acute	42	79.25	
Chronic	3	5.66	
Proposed risk factors	-	-	*p* < 0.001 (df 11)
Puerperium	28	52.83	
Pregnancy	3	5.66	
Antiphospholipid syndrome	4	7.55	
Oral contraceptive pills	7	13.21	
Pure red blood aplasia	1	1.89	
CNS infection due to sinus	2	3.77	
Leukemia	1	1.89	
Elevated protein C and S	3	5.66	
Polycythemia due to congenital heart disease	1	1.89	
Nephrotic syndrome	1	1.89	
Tuberculosis	1	1.89	
Dehydration and primary dystonia	1	1.89	

* Only age is presented as mean ± SD; ** not included in statistical analysis; df = degree of freedom.

**Table 2 diagnostics-11-00958-t002:** Comparison of demographic data, clinical characteristics, and risk factors in patients with or without parenchymal lesions on CT scan (*n* = 50).

Parameter	Patients with Parenchymal Lesions (*n* = 32)	Patients without Parenchymal Lesions (*n* = 18)	*p*-Value
	Number (%) *	Number (%) *	
Demographic data			
Age (mean ± SD) years	28.56 ± 7.86	27.77 ± 6.55	0.722
Female sex	30 (93.75)	12 (66.66)	
Marital status (married)	30 (93.75)	14 (77.77)	0.095
Clinical presentation			
Headache	30 (93.75)	17 (94.44)	0.921
Vomiting	23 (71.87)	14 (77.77)	0.648
Altered state of consciousness	19 (59.37)	9 (50.00)	0.522
Blurred vision	9 (28.12)	10 (55.55)	<0.000
Seizures	23 (71.87)	7 (38.88)	0.036
Papilledema	16 (50.00)	10 (55.55)	0.706
Weakness	16 (50.00)	9 (50.00)	-
Left side	8 (50.00)	7 (77.77)	-
Right side	8 (50.00)	2 (22.23)	-
Onset of disease			
Acute	8 (25.00)	0 (0.00)	-
Sub-acute	23 (71.87)	18 (100)	-
Chronic	1 (3.12)	0 (0.00)	-
Proposed risk factors			
Puerperium **	22/30 (93.33)	6/12 (50.00)	0.147
Pregnancy **	1/30 (3.33)	2/30 (16.66)	0.130
Antiphospholipid syndrome	2 (6.25)	2 (11.11)	0.543
Oral contraceptive pills **	4 (12.50)	1 (5.55)	0.651
Pure red blood aplasia	1 (3.12)	0 (0.00)	0.449
CNS infection due to sinus	1 (3.12)	1 (5.55)	0.674
Leukemia	1 (3.12)	0 (0.00)	0.449
Elevated protein C and S levels	0 (0.00)	2 (11.11)	0.054
Polycythemia due to congenital heart disease	0 (0.00)	1 (5.55)	0.178
Nephrotic syndrome	0 (0.00)	1 (5.55)	0.178
Tuberculosis	0 (0.00)	1 (5.55)	0.178
Dehydration and primary dystonia	0 (0.00)	1 (5.55)	0.178
Occluded sinus	47 (68.11)	22 (31.88)	<0.001
Single sinus affected	19/32 (59.37)	14/18 (77.77)	0.187
≥2 sinuses affected	13/32 (40.62)	4/18 (22.22)	0.157

* Only age is presented as mean ± SD, ** adjusted to only females (gender-specific).

**Table 3 diagnostics-11-00958-t003:** Comparison of neuroradiological findings in patients with or without parenchymal lesions on CT scan (*n* = 50).

Sinus Involved	Patients with Parenchymal Lesions (*n* = 32)	Patients without Parenchymal Lesions (*n* = 18)	*p*-Value
	Number (%)	Number (%)	
Name of sinuses affected			
Superior sagittal sinus	18 (56.25)	17 (94.44)	0.003
Left transverse sinus	13 (40.62)	1 (5.55)	0.565
Right transverse sinus	12 (37.5)	4 (22.22)	0.501
Sigmoid sinus	3 (9.37)	0 (0.00)	0.226
Straight sinus	1 (3.12)	0 (0.00)	0.491
Total	47 (68.11)	22 (31.88)	0.000
Number of sinuses involved per patient *	1.46 ± 0.621	1.22 ± 0.427	0.105
Single sinus affected (*n* = 33)			
Superior sagittal sinus	7 (36.84)	13 (39.39)	0.001
Left transverse sinus	7 (36.84)	0 (0.00)	0.011
Right transverse sinus	5 (26.31)	1 (03.03)	0.158
Total	19/33 (57.57)	14/33 (42.42)	0.218
Two sinuses affected (*n* = 15)			
Superior + left transverse	3 (27.27)	1 (25.00)	0.930
Superior + right transverse	5 (45.45)	3 (75.00)	0.310
Superior + sigmoid	1 (9.09)	0 (0.00)	0.533
Left transverse + sigmoid	1 (9.09)	0 (0.00)	0.533
Right transverse + sigmoid	1 (9.09)	0 (0.00)	0.533
Total	11/15(73.33)	4/15 (26.66)	0.011
Three sinuses affected (*n* = 2)	2 (100)	0 (0.00)	0.046
Superior + left transverse + straight	1 (50.00)	0 (0.00)	-
Superior + left transverse + right transverse	1 (50.00)	0 (0.00)	-

* Presented as mean ± SD.

## Data Availability

Requests for access to individual subject data may be made to the corresponding author by sending an email (profahsannuman@kemu.edu.pk).

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
