# Peer review of "Structural Imaging Characteristic, Clinical Features and Risk Factors of Cerebral Venous Sinus Thrombosis: A Prospective Cross-Sectional Analysis from a Tertiary Care Hospital in Pakistan"

_diagnostics, 2021, doi:10.3390/diagnostics11060958_

Round 1
Reviewer 1 Report
This manuscript describes a case series of 53 patients with sinus thrombosis in a single center in Pakistan. The major issue of the study is that is lacks novelty, however it presents interesting data regarding the population of sinus thrombosis in Pakistan. The authors present the results well and discussed appropriate differences to other populations.
- A structured abstract should be provided (introduction, material and methods, results and conclusion)
- Please provide percentage of CVT in all stroke cases in this period (in abstract as well as in the results)
- The introduction can be shortened to two paragraphs
- Please provide better image examples
- The conclusion should be modified to describe the most important findings of the study in the Pakistan population
Author Response
Dear Reviewer.
I am very much thankful for you for sparing time to review the manuscript. The suggestions/query has been addressed in a para-wise/point-to-point style. The changes has been addressed as Track Changes in the main Revised Manuscript. Once again thanks for your right suggestions for the improvement of the manuscript.
Regards
Authors

Reviewer 2 Report
Authors conducted a prospective cross-section single-center study to characterize the clinical characteristics, potential risk factors, and neuroradiological features of patients with CVST in a tertiary care hospital located in Lahore-Pakistan.
I have the following comments:
The reported casuistic is quite small especially for comparison with the larger ones cited in the paper. Caution should be used in the conclusions for this reason.
Author gave no information about management of these patients. It would be interesting to have add data on anticoagulant, anticonvulsant, anti-edema therapy for example.
There are no data about the clinical severity of clinical picture (NHISS for example) and another limit is that there are no follow-up data (for example NHISS and/or modified Rankin scale at discharge and at 3 months). Do patients recover from blurred vision?
Another aspect to consider, in order to better understand the high frequency of lesions seen on brain CT is the delay from symptom onset and CT scan. Indeed, the high frequency of altered CT together with the high frequency of symptoms expression of raised intracranial pressure (headache, seizures, vomiting, altered consciousness, papilledema) support a latency in diagnosing/presenting to the emergency Room of this population.
Minor comments:
Abstract:
Shift male to female ratio soon after 84.9 % of female frequency.
Remove the p values
Methods:
Inclusion criteria:
Authors should define how they evaluated cranial hypertension, not just remind to an article published by another group in 1997.
Inclusion criteria are not clear. Patients were included if they received a diagnosis of CVST. Authors should better describe how they diagnosed CVST. Delta sign can be seen on contrast CT not just CT. Did patients undergo CT angiography? If not, why?
Results:
Were variables normally distributed according to the Kolmogorov–Smirnov test that authors cited in the methods?
Table 1: which kind of analysis was performed? What does the p value mean?
Table 2 and table 4 are identic. Delete table 2.
Papilledema: how was it evaluated?
Lines 141-143: I would change the sentence saying that 53 patients fitted inclusion criteria….
Line 163: table 3 instead of table 2.
Line 176: what are “delta-plus cord sings”?
Line 177: delete (p<0.001)
Line 178: n=33 instead of 36
Line 180: table 2 instead of table 3
Author Response
Reviewer No 2
Authors conducted a prospective cross-section single-center study to characterize the clinical characteristics, potential risk factors, and neuroradiological features of patients with CVST in a tertiary care hospital located in Lahore-Pakistan.
I have the following comments:
Comment No 1. The reported casuistic is quite small especially for comparison with the larger ones cited in the paper. Caution should be used in the conclusions for this reason.
Reply: Thanks a lot for this suggestion. The conclusion has been rephrased accordingly and necessary statements have been added in the main text based on the results. Please See Line No. 12-16 (Page No. 9) of Conclusion Chapter.
Comment No 2. Author gave no information about management of these patients. It would be interesting to have add data on anticoagulant, anticonvulsant, anti-edema therapy for example.
Reply: Management (treatment during admission and on discharge) and followed up of patients were not discussed in this study due to few reasons i) as follow up is not completed till now due to covid- 19 (lock down as well as delayed timing for follow-up neuroimaging) ii) Patients are already on drug trial (rivaroxaban) which data will be presented later for publications.
Comment No 3. There are no data about the clinical severity of clinical picture (NHISS for example) and another limit is that there are no follow-up data (for example NHISS and/or modified Rankin scale at discharge and at 3 months). Do patients recover from blurred vision?
Reply: The present study is a part of a project that describes the occurrence of CVST and also evaluation of various treatment regimens. The present paper only covers the first phase of the project. The second phase (treatment evaluation) is still in progress and will be submitted for publication subsequently. The proposed suggestions are already covered in the follow paper. The patients with blurred visions became normal that has been discussed in the next coming publication
Comment No 4. Another aspect to consider, in order to better understand the high frequency of lesions seen on brain CT is the delay from symptom onset and CT scan. Indeed, the high frequency of altered CT together with the high frequency of symptoms expression of raised intracranial pressure (headache, seizures, vomiting, altered consciousness, papilledema) support a latency in diagnosing/presenting to the emergency Room of this population.
Reply: Thanks a lot for your suggestions. As KEMU is a tertiary hospital, therefore, we accept a number of cases that had been referred by local primary/secondary health units causing potential delay in diagnosis. This information has been incorporated in the main text. Please see Line No. 17-23 of Discussion Chapter (Page No. 8).
Minor comments:
Comment No 5.1 Abstract: Shift male to female ratio soon after 84.9 % of female frequency.
Reply: As suggested, necessary rephrasing has been carried out in the text. Please see Line No. 14-15 of Abstract Chapter
Comment No 5.2. Abstract: Remove the p values
Reply: As suggested, all p-values have been omitted from the Abstract Chapter. Please See Line No. 19, 22, 25 of Chapter Abstract.
Comment No 5.3. Methods: Inclusion criteria: Authors should define how they evaluated cranial hypertension, not just remind to an article published by another group in 1997. Inclusion criteria are not clear. Patients were included if they received a diagnosis of CVST. Authors should better describe how they diagnosed CVST. Delta sign can be seen on contrast CT not just CT. Did patients undergo CT angiography? If not, why?
Reply: Patients were recruited based on suspected clinical signs that were further diagnosed based on CT. The confirmation of CVST was carried out by MRV. Cranial hypertension was evaluated both clinically with the help of CSF/LP examination and also with neuroimaging. The information has been also added in the main text. Please see Line No 1-2 under “Study design” of Material & Methods Chapter (Page No. 3). Not all the patients were subjected to CT angiography, rather all the tentatively confirmed patients were further confirmed by Magnetic Resonance Venography (MRV).
Comment No 5.4. Results: Were variables normally distributed according to the Kolmogorov–Smirnov test that authors cited in the methods?
Reply: Kolmogorov–Smirnov test showed that all data were “Normally Distributed”. Because of normal distribution, we expressed the data as Mean± SD. Moreover, we also used Independent Student t test and Chi square after fulfilling the assumption of normal distributions of data. Generally, nature of the data regarding normal distribution is not mentioned in research papers. We also follow the same.
Comment No 5.5 Results; Table 1: which kind of analysis was performed? What does the p value mean?
Reply: In table 1, the main category of clinical presentations were subdivided into 7 variables (Headache, vomiting, altered consciousness, blurred vision, seizures, papilledema and weakness). We employed Chi-square test to know whether there is a intragroup significant difference in the frequency of occurrence of specific clinical symptoms. Seizures were more frequently found (P = 0.000) in patients, while occurrence of blurred vision were found least in CVST patients.
Similarly, Chi square was also employed for onset of disease that comprised of 3 variables (comparisons between acute, subacute and chronic) and proposed risk factors consisting of 12 variables (Puerperium, pregenancy,…….TB and Dehydration and primary dystonia). Therefore, individual p-value for each of i) Clinical presentation, ii) onset of disease and iii) proposed risk factors were mentioned in Table against each Main Category. The information has also been added in Statistical Analysis (Materials and Methods Chapter). Please line No. 5-7 (Page No. 3) of Statistical Analysis.
For further clarification, degree of freedom (df) has also been added against each of the main category in Table 1.
Comment No 5.6. Result; Table 2 and table 4 are identic. Delete table 2.
Reply: Table has been deleted
Comment No 15.7. Papilledema: how was it evaluated?
Reply: The Papilledema was diagnosed with a funduscope and was further confirmed by the qualified ophthalmologist. The information has been included in the main text. Please see Line No 3-4 under Exclusion Criteria of Material and Methods Chapter (Page No.3)
Comment No 5.8. Result; Lines 141-143: I would change the sentence saying that 53 patients fitted inclusion criteria….
Reply: As per suggestion, the sentence has been added in the main text. Please see Line No. 2 of Chapter Result (Page No. 4).
Comment No 5.9. Result;Line 163: table 3 instead of table 2.
Reply: Thanks Table No has been corrected
Comment No 5.10. Result;Line 176: what are “delta-plus cord sings”?
Reply: Sorry as it was error and correction has been made as accordingly. Please see Line No. 7 under “Neuroradiological Findings” of Result Chapter (Page No. 5).
Comment No 5.11. Result;Line 177: delete (p<0.001)
Reply: Deletion has been carried out
Comment No 5.12. Result;Line 178: n=33 instead of 36
Reply: Correction has been made
Comment No 5.13. Result;Line 180: table 2 instead of table 3
Reply: Correction about the table number has been made.
Round 2
Reviewer 2 Report
The Authors did a good job in the revision of the manuscript which improved after the changes.
1-I will add a sentence about the fact that treatments during admission and on discharge are not reported because “the present study is a part of a project that describes the occurrence of CVST and also evaluation of various treatment regimens and that in the present paper only description of basal data are reported while treatment and follow-up will be reported…
2- “delta sign” can be found on contrast cranial computed tomography and not just CT (Study design, line 103).
3-If in the methods Kolmogorov–Smirnov test is reported; the result has to be qritten in the “results” section. Otherwise, delete it from the methods.
Author Response
Reviewer No 2
The Authors did a good job in the revision of the manuscript which improved after the changes.
Thanks a lot for your kind words and I am sure that your suggestions will certainly improve the quality of the manuscript.
Comment No. 1: I will add a sentence about the fact that treatments during admission and on discharge are not reported because “the present study is a part of a project that describes the occurrence of CVST and also evaluation of various treatment regimens and that in the present paper only description of basal data are reported while treatment and follow-up will be reported…
Reply: Thanks for your suggestions. The proposed statements have been added in the main text at the end of the Discussion. Please see Line No 88-90 of Discussion Chapter (Page No. 8).
Comment No. 2- “delta sign” can be found on contrast cranial computed tomography and not just CT (Study design, line 103).
Reply: Thanks a lot. The correction has been made. Please see Line No. 103 of study Design under Material and Methods Chapter (Page No. 2).
Comment No 3-If in the methods Kolmogorov–Smirnov test is reported; the result has to be written in the “results” section. Otherwise, delete it from the methods.
Reply: The statement has been omitted from the text as suggested.